# Anti-CD20 Therapy Alters the Protein Signature in Experimental Murine AIH, but Not Exclusively towards Regeneration

**DOI:** 10.3390/cells10061471

**Published:** 2021-06-11

**Authors:** Laura Elisa Buitrago-Molina, Janine Dywicki, Fatih Noyan, Lena Schepergerdes, Julia Pietrek, Maren Lieber, Jerome Schlue, Michael P. Manns, Heiner Wedemeyer, Elmar Jaeckel, Matthias Hardtke-Wolenski

**Affiliations:** 1Department of Gastroenterology, Hepatology & Endocrinology, Hannover Medical School, 30625 Hannover, Germany; Buitrago.Laura@mh-hannover.de (L.E.B.-M.); janine.dywicki@gmx.de (J.D.); noyan.fatih@mh-hannover.de (F.N.); schepergerdes.lena@mh-hannover.de (L.S.); Lieber.Maren@mh-hannover.de (M.L.); Manns.Michael@mh-hannover.de (M.P.M.); Wedemeyer.heiner@mh-hannover.de (H.W.); jaeckel.elmar@mh-hannover.de (E.J.); 2Department of Gastroenterology and Hepatology, University Hospital Essen, University Duisburg-Essen, 45147 Essen, Germany; Julia.Pietrek@uk-essen.de; 3Institute of Pathology, Hannover Medical School, 30625 Hannover, Germany; schlue.jerome@mh-hannover.de

**Keywords:** autoimmune hepatitis, anti-CD20 therapy, immune tolerance, regeneration, hepatic inflammation

## Abstract

Background: Autoimmune hepatitis (AIH) is a chronic autoimmune inflammatory disease that usually requires lifelong immunosuppression. Frequent recurrences after the discontinuation of therapy indicate that intrahepatic immune regulation is not restored by current treatments. Studies of other autoimmune diseases suggest that temporary depletion of B cells can improve disease progression in the long term. Methods: We tested a single administration of anti-CD20 antibodies to reduce B cells and the amount of IgG to induce intrahepatic immune tolerance. We used our experimental murine AIH (emAIH) model and treated the mice with anti-CD20 during the late stage of the disease. Results: After treatment, the mice showed the expected reductions in B cells and serum IgGs, but no improvements in pathology. However, all treated animals showed a highly altered serum protein expression pattern, which was a balance between inflammation and regeneration. Conclusions: In conclusion, anti-CD20 therapy did not produce clinically measurable results because it triggered inflammation, as well as regeneration, at the proteomic level. This finding suggests that anti-CD20 is ineffective as a sole treatment for AIH or emAIH.

## 1. Introduction

Autoimmune hepatitis (AIH) is a chronic autoimmune inflammatory disease of liver tissue. AIH treatment has remained mostly the same for decades. Most patients require life-long immunosuppression and relapse after the discontinuation of therapy. The first-line treatment is corticosteroids with or without azathioprine [1,2,3]. The ideal management of nonresponders remains unclear. Current therapies block pathogenic immune responses without reestablishing immune tolerance [4]. Therefore, future therapies should aim to restore intrahepatic immune regulation to enable the discontinuation of immunosuppressive therapy. Biological treatment approaches in small cohorts have included anti-TNF-α, low-dose IL-2, and TGF-β. Another interesting approach is the use of anti-CD20 to deplete B cells and reduce the humoral immune response.

B cells and the autoantibodies they produce are highly relevant in many autoimmune diseases. However, the roles of these factors in the pathogenesis and pathophysiology of type 1 diabetes (T1D) and AIH are controversial. There is widespread agreement that both conditions are T cell-mediated autoimmune diseases. Rituximab and biosimilars (Rixathon, Truxima) are monoclonal anti-CD20 antibody therapeutics (hereafter referred to as “anti-CD20”) that deplete B cells and thus modulate the humoral immune response. These therapies are used with good success in other autoimmune diseases such as rheumatoid arthritis (RA) and systemic lupus erythematosus (SLE) [5,6,7,8].

In studies with small cohorts of patients, anti-CD20 treatment has shown success in modulating transaminases and reducing hepatic inflammation in some patients [9,10]. A small study evaluated the safety and efficacy of two doses of rituximab in five adult AIH patients who did not tolerate and did not respond to standard therapy [9]. Rituximab was found to be safe, and all patients achieved biochemical improvements after 6 months. Additionally, four of the patients who were biopsied showed decreased hepatic inflammation. Other case studies showed biochemical improvements at 3–8 months in adults and two difficult-to-treat pediatric patients [11,12,13].

Here, we used our well-established model of experimental murine AIH (emAIH) [14,15,16] and examined the effect of anti-CD20 treatment. In a series of previous studies, we demonstrated that the number of intrahepatic B cells increased after splenectomy, and the course of emAIH was more severe. A causal link between increased B cell counts and disease severity was not demonstrated. However, a temporary reduction in B cells by anti-CD20 therapy should lead to a significant improvement in this context. Therefore, we compared emAIH animals that received anti-CD20 treatment during the late course of disease with untreated controls. The histopathology, biochemical parameters, intrahepatic and intrasplenic cellular components, and activation status of the immune response were analyzed. In addition, we evaluated the signature of serum proteins that are involved in many different processes, such as angiogenesis, apoptosis, cell adhesion, differentiation, motility, proliferation, metabolic processes, chemotaxis, developmental processes, the immune response, the regulation of gene expression, and the response to stress.

## 2. Materials and Methods

### 2.1. Mice

Animals were maintained under specific pathogen-free conditions at the Central Animal Facility of Hannover Medical School (Hannover, Germany). NOD/Ltj mice were intravenously injected with a total of 4 × 10^9^ infectious particles containing adenovirus (Ad)-FTCD (formiminotransferase cyclodeaminase) in PBS [14,15,16]. Six of the animals were randomly injected with 250 µg of anti-CD20 (Bio-X-Cell) i.v. once at week 10. All mice were sacrificed 12 weeks postinfection.

### 2.2. Adenovirus Construction

The generation of Ad-FTCD has been previously described [14,15,16]. Briefly, FTCD was amplified by PCR from cDNA generated from human liver cells; the sequence was verified by sequencing both DNA strands. The constructs were cloned into the Ad transfer vector pShuttle-CMV (Stratagene, Waldbronn, Germany). By homologous recombination, this shuttle vector was recombined with pAdEasy-1, which carried deletions in the E1 and E3 regions. The genome of the generated adenovirus could be amplified only within the HEK 293 packaging cell line, which complements the essential regions. The purification of recombinant adenovirus was performed using a cesium chloride gradient, and the adenoviral stocks were quantified using an Adeno-X™ rapid titer kit (Clontech, Saint-Germain-en-Laye, France).

### 2.3. Histology and Immunohistology

Murine livers were fixed in formalin and embedded in paraffin. Paraffin-embedded sections (5 µm) were prepared for hematoxylin and eosin (HE) staining. After being stained, the sections were examined in a blinded manner by a pathologist using the approved modified hepatitis activity index (mHAI) for autoimmune hepatitis that consists of: (A) periportal or periseptal interface hepatitis (piecemeal necrosis), (B) confluent necrosis, (C) focal (spotty) lytic necrosis, apoptosis, and focal inflammation, and (D) portal inflammation [1]. In addition, the area of the infiltrate in µm^2^ was measured using the Axiovision software (Zeiss, Jena, Deutschland).

Indirect immunofluorescence staining for the detection of autoantibodies was performed on rat sections of stomach, kidney, and liver, and slides with HepG2 cells normally used for diagnostics, kindly provided by Stephanie Loges and the Laboratory for detection of liver-specific autoantibodies at the Department of Gastroenterology, Hepatology, and Endocrinology, Hannover Medical School, Hannover, Germany. Sections were blocked with goat serum (Sigma, Hamburg-Hausbruch, Germany) in TBS-T buffer. Sera of mice were diluted 1:80. 1:160, 1:320, and 1:640 in TBS-T buffer and detected with anti-mouse-DyLight488 (Jackson Immunoresearch, Cambridgeshire, United Kingdom). All images were prepared with an AxioImagerM1 (Zeiss, Jena, Deutschland) using Axiovision software.

### 2.4. Flow Cytometry

After grinding the fresh organs, intrahepatic lymphocytes (IHLs) were separated using a 40%/70% Percoll (GE Healthcare, München, Germany) gradient. Red blood cells in the spleen were lysed, and lymphocytes were subsequently stained with appropriate combinations of anti-CD3, anti-CD45R, anti-CD4, anti-CD8, anti-CD44, anti-CD62L, anti-Foxp3, and anti-Ki-67 (all BioLegend). All data were acquired with an LSRII SORP interfaced with FACSDiva software (BD Biosciences, Heidelberg, Germany).

### 2.5. Serum Analysis

Blood samples were collected via the retro-orbital route. Serum aspartate aminotransferase (AST) and alanine transaminase (ALT) were measured by photometric enzyme activity assays with an Olympus AU400 chemistry analyzer as previously described [14,15,16].

### 2.6. Serum Protein Analysis by Olink

Proteins were measured using the Olink^®^ Mouse Exploratory Panel* (Olink Proteomics AB, Uppsala, Sweden) according to the manufacturer’s instructions as described previously [17]. The proximity extension assay (PEA) technology used for the Olink protocol has been well described [18] and allows 92 analytes to be analyzed simultaneously. Briefly, pairs of oligonucleotide-labeled antibody probes bind to their targeted protein, and if the 2 probes are brought in close proximity, the oligonucleotides will hybridize in a pairwise manner. The addition of DNA polymerase leads to a proximity-dependent DNA polymerization event, which generates a unique PCR target sequence. The resulting DNA sequence was subsequently detected and quantified using a microfluidic real-time PCR instrument (Biomark HD, Fluidigm, München, Germany) (Figure 6A). The data are then quality controlled and normalized using an internal extension control and an interplate control to adjust for intra- and interrun variations. The final assay read-out is presented in Normalized Protein eXpression (NPX) values, arbitrary units on a log2-scale in which a high value corresponds to high protein expression. All assay validation data (detection limits, intra- and interassay precision data, etc.) are available on the manufacturer’s website (www.olink.com; accessed date 1 May 2021).

### 2.7. Statistical Analysis

Unpaired, 2-tailed Student’s t-tests were performed using GraphPad Prism version 7.00 for Mac (GraphPad Software, La Jolla, CA, USA, www.graphpad.com, accessed date 1 May 2021). For Figure 5, Welch’s t-test had to be used because equal SDs could not be assumed. Alternatively, for multiple variables, the unpaired Student’s 2-tailed t-test with an implemented Benjamini-Hochberg multiplicity correction was performed using Qlucore Omics Explorer software 3.5 (Qlucore, Lund, Sweden). The heatmap represents the multiple protein expression profiles (*p* < 0.05; q < 0.05). Significant differences with *p* ≤ 0.05 are indicated by *, very significant differences (*p* ≤ 0.01) by **, and extremely significant by *** (*p* ≤ 0.001). *p* > 0.05 was considered to be not significant.

## 3. Results

### 3.1. The Role of B Cells in the Pathophysiology of emAIH

We have already characterized our emAIH model very thoroughly. In brief, FTCD was amplified from human liver cells and cloned into the Ad transfer vector. By homologous recombination, this shuttle vector was recombined with pAdEasy-1, which carried deletions in the E1 and E3 regions (Figure 1A). Nonreplicating adenoviral expression of the FTCD protein was observed in the livers of NOD/Ltj mice [16]. Twelve weeks after induction, there is intrahepatic inflammation, elevated transaminases, and massive amounts of autoantibodies in 91% of animals [16]. Other mouse strains or other known autoantigens do not elicit disease except for autoantibodies [15]. The number of B cells was neither intrasplenically nor intrahepatically increased, whereas IgG levels were strongly elevated in emAIH animals, similar to human AIH [16]. In the present model, we administered 250 µg of anti-CD20 antibody once at week 10 after emAIH induction (Figure 1B). This corresponds to the administration of the human drug rituximab, which is used to treat some autoimmune diseases and non-Hodgkin’s lymphomas.

### 3.2. Inflammation Severity Correlates with the Amount of Autoantibodies

Immunoglobulins found on the surface of hepatocytes from AIH liver biopsies facilitated antibody-mediated cytotoxicity [19]. However, the transfer of serum from emAIH mice alone did not induce pathogenesis in NODscid mice in our model (Figure 2A). As described, most of the Ad-FTCD induced either anti-nuclear autoantibodies (ANAs) or a 5.5-fold increase in anti-cytosolic autoantibodies (ACAs) (Figure 2B) [16]. We classified autoantibodies into two clinically relevant groups: those that provided positive signals on hepatic rat cryosections and HepG2 cells at a 1/80 dilution and those that did so at a 1/320 dilution. We correlated these observations with the degree of inflammation by measuring intrahepatic infiltrates (Figure 2C,D). The transfer of serum from emAIH mice and thus also of autoantibodies did not induce pathogenesis in NODscid. Therefore, the shown high and significant correlation of autoantibodies and inflammation was contrary to our expectations. This coincidence suggested causality.

### 3.3. Anti-CD20 Treatment Reduced the Number of B Cells and Subsequently the Amount of Immunoglobulin in the Sera

In mice, there are two methods to deplete B lymphocytes, either with injection of anti-CD19 or with anti-CD20. To stay within a relevant system that was comparable to humans, we used anti-CD20, which is used in humans as the drug rituximab, among others. A single administration of 250 µg at week 10 after emAIH induction resulted in a strong reduction in B cells in the spleen and liver after 20 days (Figure 3A). Surprisingly, this large reduction did not lead to a sustained decrease in the total number of lymphocytes in the liver but only in the spleen (Figure 3B). In parallel, there was a greater than 25% reduction in serum immunoglobulin concentrations in anti-CD20-treated mice (Figure 3C). Thus, the efficacy and effectiveness of depleting CD20^+^ cells were well demonstrated. However, the most important question was whether the condition of the mice with emAIH improved clinically/pathologically or biochemically.

### 3.4. There Is No Indication of Anti-CD20 Treatment Improving Pathology or Lowering Transaminases

Various treatments, such as prednisolone, budesonide, and complexed IL-2, have produced significant improvements in pathology and/or transaminases in the emAIH model [16,20]. Others have shown positive effects of anti-CD20 in their animal models [21]. Measurable parameters for AIH therapy are histological analysis and biochemical factors, especially the analysis of transaminases. We have already shown in previous studies that the mHAI in mice was significantly lower than that in AIH patients [16,22]. To our surprise, there were no beneficial effects on histopathology (Figure 4A,B). The mHAI, rated as an average of 3, was composed of observed periportal interface hepatitis, apoptosis, and portal inflammation, whereas confluent necrosis was absent in both groups. Additionally, measurement of the infiltrate sizes did not reveal any differences (data not shown). While the level of liver-specific ALT was not altered, there was a slight increasing trend in AST after anti-CD20 treatment (Figure 4C). This deterioration, albeit slight, made us very suspicious and suggested the need for further study.

### 3.5. B Cell Depletion Did Not Affect T Cell Compartments

Although the role of B cells and autoantibodies remains debatable, the role of T cells is mostly undisputed. We have shown in this model that CD4^+^ T cells are the drivers of emAIH [16]. Therefore, the roles of regulatory T cells and CD4^+^ T effectors are particularly interesting. While the number of B cells was greatly reduced by anti-CD20 administration, the total number of CD3^+^ T cells remained largely unchanged in the liver and spleen (Figure 5A,B). We demonstrated that the intrahepatic ratio of CD4^+^/CD8^+^ T cells was reduced with increasing severity of AIH in patients [4]. However, analysis of this parameter did not show any change in the liver or the spleen (Figure 5C,D). The number of regulators (CD3^+^CD4^+^CD25^+^Foxp3^+^) and potentially autoreactive T effector cells (CD3^+^CD4^+^CD44^+^CD62L^−^) also remained constant both in the liver and spleen, which represented systemic levels (Figure 5E–H). We also analyzed the activation marker CD25 on non-Tregs and the proliferation marker Ki-67 and found no differences (data not shown). In summary, T cells did not tend to expand or become more activated as a result of anti-CD20 treatment, which reduces the concern about the slightly elevated AST levels.

### 3.6. Anti-CD20 Therapy Induced a Balance between Inflammation and Regeneration at the Proteomic Level

Alarmed by the elevated AST levels, we also undertook a large-scale investigation of many inflammatory and immunologic parameters. We used the OLINK platform to analyze 92 serum proteins by PEA. In brief, a pair of oligonucleotide-labeled antibodies bound to their target protein, and if the pair was brought in close proximity, the oligonucleotides hybridized in a pairwise manner. DNA polymerase generated a unique PCR target sequence. The subsequent DNA sequence was detected and quantified by real-time PCR. Data values represent the Normalized Protein eXpression (NPX) and were analyzed using algorithms in Qlucore (Figure 6A). All analyzed proteins are associated with angiogenesis, apoptotic processes, cell adhesion, cell differentiation, cell motility, cell proliferation, cellular metabolic processes, chemotaxis, developmental processes, immune response, MAPK cascade, neurogenesis, regulation of gene expression, response to stress, and signal transduction. To show the clear groupings of the anti-CD20-treated and untreated control groups, t-distributed stochastic neighbor embedding (t-SNE) was used (Figure 6B). In total, 11/92 proteins were differentially expressed (Figure 6C; *p* < 0.006; q < 0.05). Representative examples of upregulated proteins were follistatin (FST), interleukin-5 (IL-5), and peroxiredoxin-5 (Prdx-5), whereas contactin-1 (Cntn-1) and WNT1-inducible-signaling pathway protein 1 (WISP-1) were downregulated after anti-CD20 administration as compared to untreated animals with emAIH (Figure 6D). Figure 6E provides a summary of which proteins were associated with regeneration or inflammation and their expression status (Figure 6E). Taken together, these results suggest that anti-CD20 treatment induced not only regeneration but also inflammation at the molecular level.

## 4. Discussion

In our emAIH model, we were able to show the effects of anti-CD20 therapy at the cellular and molecular levels. While the disease severity was unaffected by anti-CD20 treatment, serum IgG was clearly reduced. Additionally, there was the expected reduction in the B cells. More significant findings were found at the molecular level. Proinflammatory and regenerative proteins were balanced in this model, and therefore no disease improvement occurred.

AIH requires life-long immunosuppression by steroids with or without azathioprine in 70–80% of patients. This therapy depletes intrahepatic Tregs more efficiently than Teffs, which may explain the high relapse rate (greater than 70%) after the discontinuation of therapy [4]. Therefore, restoring local intrahepatic immune regulation may enable patients to discontinue immune suppression in the future. In this model, intrahepatic tolerance should be restored by B cell depletion using anti-CD20 administration. The rationale for this was because after splenectomy in our emAIH model, the number of intrahepatic B cells was greatly increased and the disease worsened [14,17].

Anti-CD20 therapy has been used as a drug to treat non-Hodgkin’s lymphoma and chronic lymphocytic leukemia [23,24,25]. Later, it was used as treatment for RA [26,27]. Furthermore, rituximab is frequently used off-label to treat difficult cases of multiple sclerosis (MS), SLE, chronic inflammatory demyelinating polyneuropathy (CIPD), and autoimmune anemias [5,6,7,8,28,29,30,31]. In a xenoimmunized mouse model of autoimmune hepatitis, treatment with anti-CD20 at the peak of disease showed clear positive effects on the manifestation and progression of hepatitis, as represented by reductions in the mHAI and ALT [21]. Here, we did not observe a reduction in transaminases or the mHAI, but the model was not comparable since there were different mouse strains, different induction protocols, and different courses of disease. Nonetheless, both models demonstrated the positive effect of a single anti-CD20 treatment.

It has been shown that rituximab induced IL-6 in human B cells [32]. In our model, no regulation of IL-6 was observed. However, after anti-CD20 administration, inflammatory IL-5 stimulated B cell growth and increased immunoglobulin secretion (Figure 6C–E). Both outcomes could be expected after B cell depletion. However, IL-5 is known to prolong inflammation, unlike a regenerative cytokine [33]. Likewise, upregulation of proinflammatory Ccl20 is a clear signal to enhance the T cell response [34,35]. In certain exceptional conditions, this signal may also affect the Treg population [36,37], but this effect was not observed in the present model. Rather, an increased level of Ccl20 in MS is a clear signal of an increase in inflammatory events and thus worsening of the disease [38]. In addition, others have already shown that rituximab may induce Ccl20 expression in human blood [39]. Furthermore, the expression of Fst is associated with liver fibrosis [40,41], while the downregulation of Clstn2 was observed in progressive RA [42]. At this point, it is reasonable to speculate about a connection between the cytokine release syndrome and rituximab, which is certainly known in certain circumstances [43].

To our knowledge, the positive effects of anti-CD20 via the expression of proteins such as Prdx-5 or the modulation of Cntn-1 and WISP-1 have not been shown thus far. Prdx-5 has antioxidative and cytoprotective effects during oxidative stress [44,45]. Therefore, upregulation of this factor might be one mechanism of the positive effects observed in other models [21] and diseases [5,6,7,8,9,10,11,12,13]. The same is true for the modulation of WISP-1, which is upregulated in patients and mice with fibrosis [46,47], and overexpression of WISP-1 promotes tumor growth [48]. The elusive role of Pdgfb in fibrotic diseases and systemic sclerosis has been shown [49]. Therefore, Pdgfb downregulation is a useful approach in disease control and regeneration. The effect of anti-CD20 administration on Cntn-1 is very interesting. On the one hand, upregulated Cntn-1 is a marker of poor outcome in tumor diseases [50,51]; on the other hand, autoantibodies against Cntn-1 are disease-promoting in CIPD [52]. CIPD is an autoinflammatory disease for which rituximab is successfully used [8]. From this finding, it can be hypothesized that the positive effect of rituximab on CIPD may result precisely from this effect. First, Cntn-1 was reduced, and then, the targets of autoantibodies were reduced.

In autoimmune diseases such as AIH, strengthening intrahepatic immune regulation seems to be an obvious strategy. Rituximab has been shown to be effective in some autoimmune diseases, so the question remains as to why it is neither clinically effective in the treatment of AIH nor in the emAIH model. Among the molecules and presented mechanisms, IL-17 was shown to be expressed after rituximab administration [53]. Therefore, we suggest that anti-CD20 therapy should be combined with complexed, low-dose anti-IL-2 therapy, adoptive Treg transfer, or both [20,54]. The concept of low-dose IL-2 therapy to treat autoimmunity was suggested because the high-affinity IL-2 receptor is known to be constitutively expressed on Tregs [55]. Both Tregs and IL-2 are essential components for the maintenance of immune tolerance. Low-dose IL-2 increases the number of Tregs; it has clinically used for the treatment of lupus and lupus nephritis [56,57] and is currently being tested in AIH [58,59] and other autoimmune diseases [60,61].

## 5. Conclusions

In conclusion, anti-CD20 treatment in AIH and emAIH remains an interesting tool for the treatment of the disease but is obviously not suitable for monotherapy. We showed that this effect was due to the consequences of treatment at the protein level. Here, too many proinflammatory proteins were induced to allow regeneration. Therefore, other immunological treatments must be considered for reestablishing immune tolerance in patients with AIH.

## Figures and Tables

**Figure 1 cells-10-01471-f001:**
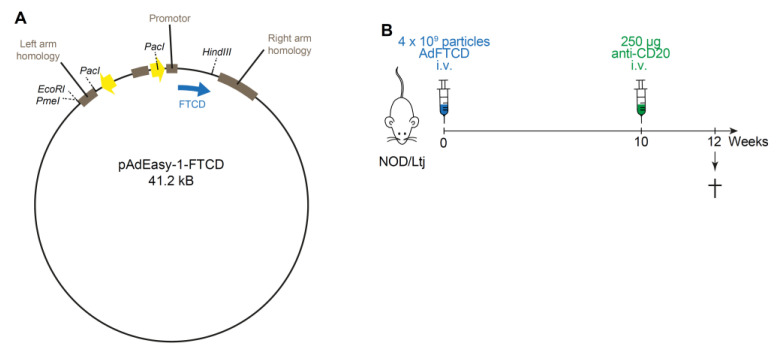
Schematic overview of the Ad-FTCD construct and the experimental setup. (**A**) Scheme of 41.2 kB adenoviral construct coding for FTCD with the most important cloning sites. (**B**) Experimental scheme of emAIH induction and anti-CD20 treatment in NOD/Ltj mice.

**Figure 2 cells-10-01471-f002:**
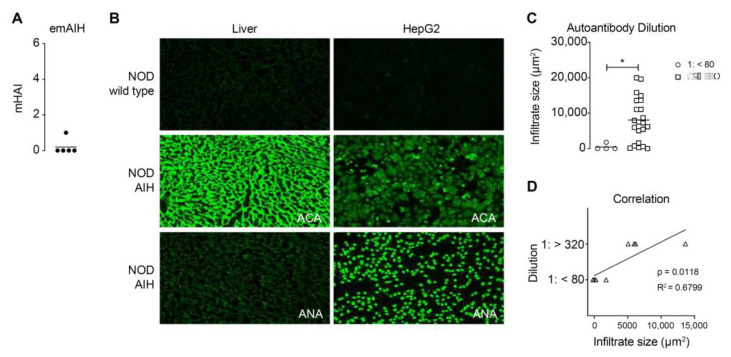
Inflammation severity correlates with the number of autoantibodies. (**A**) Modified hepatitis activity index of NODscid mice that received sera of emAIH-bearing NOD/Ltj mice. (**B**) Examples of indirect immunofluorescence of rat liver sections or HepG2 cells with sera of wildtype NOD/Ltj (controls, upper row) and Ad-hFTCD-infected emAIH-bearing NOD/Ltj mice (middle and lower row). (**C**) Quantification of sera from emAIH bearing NOD/Ltj giving positive signals in (**B**) in a dilution of less than 1:80 (left group) or more than 1:320 (right group) in comparison to the size of hepatic lymphocyte infiltrates (* *p* ≤ 0.05; *n* = 26). (**D**) Correlation of the infiltrate size and the dilution of emAIH sera containing autoantibodies.

**Figure 3 cells-10-01471-f003:**
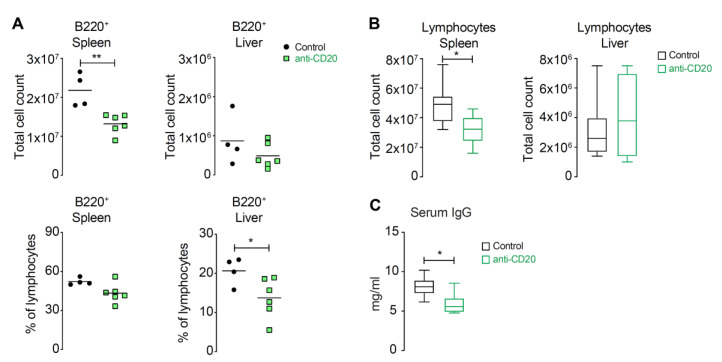
Anti-CD20 treatment reduced the number of B cells and subsequently the amount of immunoglobulin in the sera. (**A**) Anti-CD20-treated (anti-CD20; *n* = 6) and untreated NOD/Ltj control mice with emAIH (control; *n* = 4) were analyzed 20 days after anti-CD20 treatment to determine the intrahepatic frequencies of B220^+^CD3^-^ (B220^+^) cells in spleen (left graphs) or liver (right graphs). Upper graphs show total cell numbers, while lower graphs show B220^+^ cells among splenocytes or IHLs by flow cytometry. (* *p* ≤ 0.05; ** *p* ≤ 0.01) (**B**) Total splenocyte numbers (left graph) or IHLs (right graph) of anti-CD20-treated (anti-CD20) and untreated NOD/Ltj control mice with emAIH (control) at the same timepoint. (* *p* ≤ 0.05) (**C**) Total serum IgG of anti-CD20-treated (anti-CD20) and untreated NOD/Ltj control mice with emAIH (control) was determined 20 days after anti-CD20 treatment (* *p* ≤ 0.05).

**Figure 4 cells-10-01471-f004:**
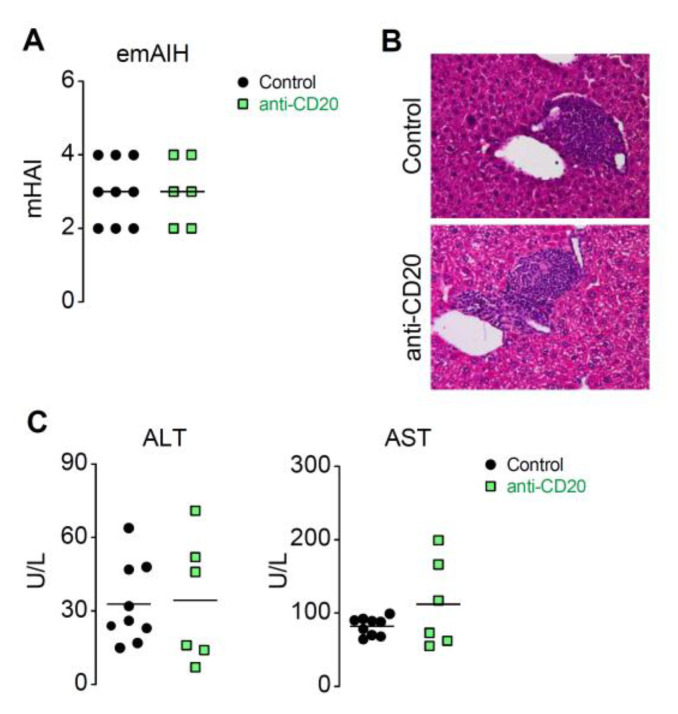
There is no indication of anti-CD20 treatment improving pathology or lowering transaminases. (**A**) 20 days after treatment of hepatic sections the histological mHAI was taken. (**B**) Microscopic HE staining. (**C**) The serum ALT and AST levels were measured after anti-CD20 therapy initiation in treated mice (anti-CD20; *n* = 6) and controls not given therapy (control; *n* = 9).

**Figure 5 cells-10-01471-f005:**
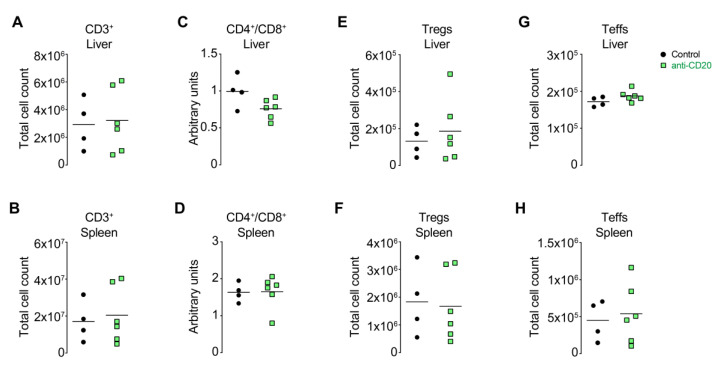
B cell depletion did not affect T cell compartments. Anti-CD20-treated (anti-CD20; *n* = 6) and untreated NOD/Ltj control mice with emAIH (control; *n* = 4) were analyzed 20 days after anti-CD20 treatment for (**A**) the quantification of total CD3^+^ T cells in the liver or (**B**) spleen. (**C**) Ratio of CD3^+^CD4^+^/CD3^+^CD8^+^ in the liver or (**D**) spleen. (**E**) Quantification of total CD3^+^CD4^+^CD25^+^Foxp3^+^ Tregs in the liver or (**F**) spleen and (**G**) CD3^+^CD4^+^ and CD3^+^CD8^+^ effector T cells (Teffs) in the liver and (**H**) spleen.

**Figure 6 cells-10-01471-f006:**
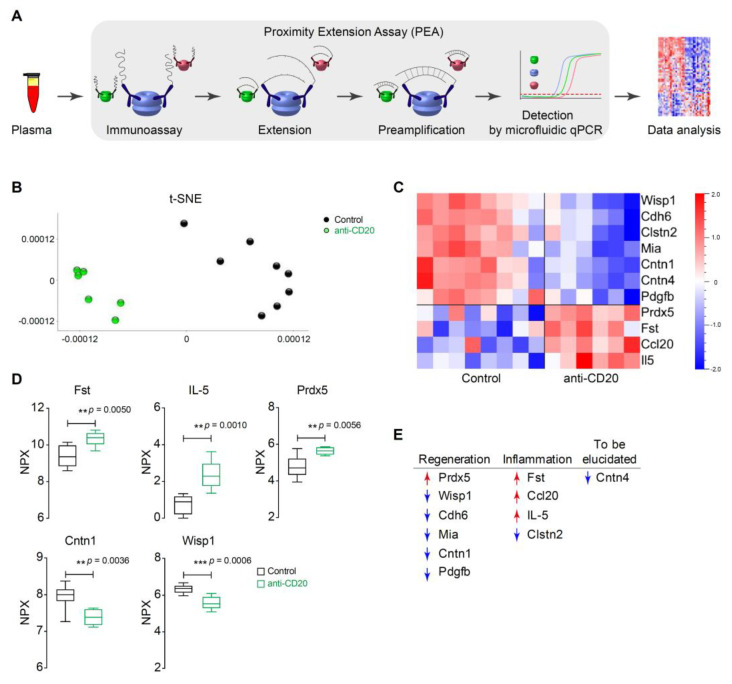
Anti-CD20 therapy induced a balance between inflammation and regeneration at the proteomic level. (**A**) Schema of the proximity extension assay to quantify serum proteins. (**B**) The t-SNE visualizes high-dimensional data by giving each mice a location in a map. (**C**) The heatmap shows serum analyses of anti-CD20-treated (*n* = 6) and control NOD/Ltj mice (*n* = 8) for 92 proteins by Olink technology. The heatmaps shown take into account the multiplicity correction after calculating the *p* (< 0.05) and q values (< 0.05) for all 92 proteins. (**D**) Single analysis of the selected protein Fst, IL-5, Prdx-5, Cntn-1, and WISP-1 obtained from (**A**). (**E**) The 11 significant proteins from (**C**) were plotted in a table related to their function in accordance to their regulation status.

## Data Availability

The data presented in this study are available on request from the corresponding author.

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
