# Peer review of "Anti-CD20 Therapy Alters the Protein Signature in Experimental Murine AIH, but Not Exclusively towards Regeneration"

_cells, 2021, doi:10.3390/cells10061471_

Round 1

Reviewer 1 Report

The paper entitled "Anti-CD20 alters the protein signature in experimental murine AIH but not exclusively towards regeneration" is a very elegant experimental work aiming to assess the role of anti-CD20 treatment in Autoimmune Hepatitis (AIH). This represents a quite hot topic in the everyday clinical practice especially in the case of patients with severe AIH or nonresponders to classical therapy .  The results of this work show that anti CD-20 reduces the IgG level and the amount of B cells in the tissue but interestingly it has no effect on disease severity (ALT level, histology). I believe that this paper really contribute to the improvement of the knowledge in this field and it should be published without major changes. I have only one  question to the authors reading this paper. Is it possible that the negative results of the anti-CD20 treatment may be just simply "time related" or "dose related"? If you have had sacrificed the mice later (for example 4 or 6 weeks after the treatment intervention and not at week 2) do you believe that you could  have observed some different results regarding histology ??? In the few case reports an improvement in the patients was observed much later (months) after the treatment with Rituximab.   A comment about this in the discussion could be interesting. 

Author Response

Reviewer 1: The paper entitled "Anti-CD20 alters the protein signature in experimental murine AIH but not exclusively towards regeneration" is a very elegant experimental work aiming to assess the role of anti-CD20 treatment in Autoimmune Hepatitis (AIH). This represents a quite hot topic in the everyday clinical practice especially in the case of patients with severe AIH or nonresponders to classical therapy .  The results of this work show that anti CD-20 reduces the IgG level and the amount of B cells in the tissue but interestingly it has no effect on disease severity (ALT level, histology). I believe that this paper really contribute to the improvement of the knowledge in this field and it should be published without major changes.

We thank Reviewer 1 for the effort and expense, and for sharing our assessment that the manuscript is “quite hot topic in the everyday clinical practice”.

Reviewer 1:I have only one  question to the authors reading this paper. Is it possible that the negative results of the anti-CD20 treatment may be just simply "time related" or "dose related"? If you have had sacrificed the mice later (for example 4 or 6 weeks after the treatment intervention and not at week 2) do you believe that you could have observed some different results regarding histology ??? In the few case reports an improvement in the patients was observed much later (months) after the treatment with Rituximab.   A comment about this in the discussion could be interesting. 

We totally agree with the reviewer 1 that dosage and the time between treatment and analysis might play a role. For the question of dose finding, we were guided on the one hand by the literature, and on the other hand by the effect on the depletion of B cells. We think that the dose is optimally chosen, since a low dose would have depleted significantly less B cells, while a higher dose would probably have increased the risk of cytokine release syndrome. In accordance with the literature, where others have shown positive effects of the same concentration of anti-CD20 in their animal model [1], we also used 250 µg /mouse.

The second half of the question is a little harder to answer. Our experience with prednisolone, budesonide [2] and IL-2/anti-2 complex [3], among others, has shown that positive effects can be clearly visible even after two weeks. This applies to histological improvements as well as to transaminases. However, it cannot be definitively ruled out. But, this is a problem in the emAIH model. The animals used are NOD/Ltj. These are known to the scientific community as the T1D model. If the model is run for much longer than 12 weeks after induction, animals also develop spontaneously T1D [2, 4]. In animals that do not develop T1D, we observed a progressive deterioration that is also associated with fibrosis [2]. However, due to the severity of the disease, we lose about 90% of the experimental animals and would therefore possibly cause a bias by the anti-CD20 therapy if there is a positive effect on the development of T1D. In summary, we doubt that the positive effect of anti-CD20 would only occur with a time delay in the emAIH model.

References:

  1. Beland, K., et al., Depletion of B cells induces remission of autoimmune hepatitis in mice through reduced antigen presentation and help to T cells. Hepatology, 2015. 62(5): p. 1511-23.
  2. Hardtke-Wolenski, M., et al., Genetic predisposition and environmental danger signals initiate chronic autoimmune hepatitis driven by CD4+ T cells. Hepatology, 2013. 58(2): p. 718-28.
  3. Buitrago-Molina, L.E., et al., Treg-specific IL-2 therapy can reestablish intrahepatic immune regulation in autoimmune hepatitis. J Autoimmun, 2021. 117: p. 102591.
  4. Hardtke-Wolenski, M., et al., The influence of genetic predisposition and autoimmune hepatitis inducing antigens in disease development. J Autoimmun, 2017. 78: p. 39-45.

Reviewer 2 Report

eviewer’s comments

This original article primarily focused on the efficacy of anti-CD20 in the experimental animal model of AIH. It is of interest that the treatment with anti-CD20 kept a balance between regeneration and inflammation in the experimental animal model of AIH. However, experimental designs were inappropriate in some parts. In addition, the interpretations of results obtained by this study seem to be insufficient. The putative mechanisms by which anti-CD20 acted were not fully described. It is regrettable to say that this article is not acceptable for publication. Please refer to the comments shown below.

Major

#1. The authors should describe how the dose and timing of anti-CD 20 were determined. This study requires a preliminary experiment to determine it. The optimal dose and timing of anti-CD 20 is extremely important to acquire its maximal effects.

#2. The methods for detecting ANA and AMA should be described in “Materials and Methods”.

Rat kidney/stomach is usually used as a substrate for an indirect immunofluorescence method.

But, rat liver was used for detecting AMA (Figure.2 middle). Also, AMA was detecting in the animal model of AIH. It seemed to be strange for the animal model of AIH.

#3. Intrahepatic lymphocytes were assayed, using anti-Fox3, anti-Ki67 and so on (Lines 96-97). But, the results are missing.

#4. The definition of “infiltrate size” should be described (Figure 2D). Sample numbers (n=**) should be addressed in the figure.

#5. The authors should describe the factors which the modified HAI contains.

#6. The data in wild type should be added as a comparison group (Figure 3). In addition, titers of ANA should be determined in control and anti-CD20 groups.

#7. Figure 3B showed the decrease in the number of lymphocytes was observed in the spleen, but not in the liver. The authors should describe the reason.

#8. Histological findings, including inflammatory cells infiltration, and hepatic fibrosis, should be described more in detail (Figure 4B).

#9. The fold change and p-value in each protein should be addressed (Figure 6). In Figure 6D, Wisp1 expression, which is up-regulated in the mice with fibrosis, was significantly decreased by the treatment with anti-CD20. But, no significant histological change was not found by the treatment. Moreover, IL-5 expression was enhanced by the treatment. The authors should mention the reasons. Tthe roles of Fst and Cntn1 should be also described more in detail.

#10. I found the statement, “In our model, no regulation of IL-6 was observed (Line 291-292)”. But, its result was not shown in this study.

#11. This study revealed that the treatment with anti-CD20 kept a balance between regeneration and inflammation in the model of AIH. But, the authors should describe how the treatment kept the balance more clearly.

Minor

#1. “At week ten” should be corrected to “at ten week” (line 140).

#2. “Fig C,D.” should be corrected to “Fig.2C,D” (line 156).

#3. Sample number of the control group (n=4) in Figure 5 should be addressed.

#4. The authors need to explain “q<0.05” (line 247).

#5. “On the other hand” is duplicated (Lines 312-313).

Author Response

Reviewer 2: This original article primarily focused on the efficacy of anti-CD20 in the experimental animal model of AIH. It is of interest that the treatment with anti-CD20 kept a balance between regeneration and inflammation in the experimental animal model of AIH. However, experimental designs were inappropriate in some parts. In addition, the interpretations of results obtained by this study seem to be insufficient. The putative mechanisms by which anti-CD20 acted were not fully described. It is regrettable to say that this article is not acceptable for publication. Please refer to the comments shown below.

We regret that we could not fill Reviewer 2 with enthusiasm about the original version of the manuscript. In the following, we will try to convince him of the correctness of our experiments, presentations and conclusions. Anyways, we thank Reviewer 3 for the effort and suggestions for improvement.

Reviewer 2: #1. The authors should describe how the dose and timing of anti-CD20 were determined. This study requires a preliminary experiment to determine it. The optimal dose and timing of anti-CD20 is extremely important to acquire its maximal effects.

Reviewer 2 brings up the same question here that Reviewer 1 noted. The answer is therefore the same and we will answer with copy-paste:

We totally agree with the reviewer 2 that dosage and the time between treatment and analysis might play a role. For the question of dose finding, we were guided on the one hand by the literature, and on the other hand by the effect on the depletion of B cells. We think that the dose is optimally chosen, since a low dose would have depleted significantly less B cells, while a higher dose would probably have increased the risk of cytokine release syndrome. In accordance with the literature, where others have shown positive effects of the same concentration of anti-CD20 in their animal model [1], we also used 250 µg /mouse.

The second half of the question is a little harder to answer. Our experience with prednisolone, budesonide [2] and IL-2/anti-2 complex [3], among others, has shown that positive effects can be clearly visible even after two weeks. This applies to histological improvements as well as to transaminases. However, it cannot be definitively ruled out. But, this is a problem in the emAIH model. The animals used are NOD/Ltj. These are known to the scientific community as the T1D model. If the model is run for much longer than 12 weeks after induction, animals also develop spontaneously T1D [2, 4]. In animals that do not develop T1D, we observed a progressive deterioration that is also associated with fibrosis [2]. However, due to the severity of the disease, we lose about 90% of the experimental animals and would therefore possibly cause a bias by the anti-CD20 therapy if there is a positive effect on the development of T1D. In summary, we doubt that the positive effect of anti-CD20 would only occur with a time delay in the emAIH model.

Reviewer 2: #2. The methods for detecting ANA and AMA should be described in “Materials and Methods”.

Rat kidney/stomach is usually used as a substrate for an indirect immunofluorescence method.

But, rat liver was used for detecting AMA (Figure.2 middle). Also, AMA was detecting in the animal model of AIH. It seemed to be strange for the animal model of AIH.

We totally agree that the indirect immunofluorescence staining should be added to the methods section and did so.

Furthermore, we are very grateful to reviewer 2 for pointing out this error. Of course, we did not detect AMAs in our model, but anti-cytosolic antibodies. We have improved this line (line 183), replaced the Fig. 2 and would like to thank reviewer 2 and apologize.

Reviewer 2: #3. Intrahepatic lymphocytes were assayed, using anti-Fox3, anti-Ki67 and so on (Lines 96-97). But, the results are missing.

We measured anti-CD3, anti-CD45R, anti-CD4, anti-CD8, anti-CD44, anti-CD62L, anti-Foxp3 and anti-Ki-67 in flow-cytometry. In the last version we removed the figures for Ki-67 because no differences could be seen, but still mentioned the measurement in the text (line 264). The Foxp3 staining is part of “Figure 5” as well as all other markers.

Reviewer 2: #4. The definition of “infiltrate size” should be described (Figure 2D). Sample numbers (n=**) should be addressed in the figure.

The infiltrate size is the area of the infiltrate in µm2 measured by Axiovision software (Zeiss). We added this in the method section (line 103-104).

We added the “n=26” in the figure legend.

Reviewer 2: #5. The authors should describe the factors which the modified HAI contains.

The modified hepatitis activity index consists of (A) Periportal or Periseptal Interface Hepatitis (piecemeal necrosis) (B) Confluent Necrosis (C) Focal (spotty) Lytic Necrosis, Apoptosis, and Focal Inflammation (D) Portal Inflammation. We added this in the method section and cited the AIH guidelines (line 100-103).

Reviewer 2: #6. The data in wild type should be added as a comparison group (Figure 3). In addition, titers of ANA should be determined in control and anti-CD20 groups.

We are not quite sure what reviewer 2 means exactly. Figure 3 consists of flow-cytometry data, Figure 2 shows the autoantibodies.

To considered both possibilities: We have shown and published that there is no auto-Ak in wild-type mice [2, 4] and there is no change in cellular composition in flow cytometry between wild-type and emAIH [2]. Therefore, this is not a control that we were allowed to perform again in accordance with EU-wide or national animal welfare regulations.

In additional, the total amount of IgG was reduced by “just” 25% after anti-CD20 treatment. Therefore, a reduction of ANAs after two weeks is very unlikely. Furthermore, the group size is too small to be able to make a reliable statement on this question. For this, the experimental setup would have had to be different.

Reviewer 2: #7. Figure 3B showed the decrease in the number of lymphocytes was observed in the spleen, but not in the liver. The authors should describe the reason.

We agree with reviewer 2 that the finding is very interesting, but as we mentioned in the manuscript, this was surprising for us as well. We did not find any similar in the literature. Nonetheless, this was a descriptive observation. Therefore, unfortunately, we cannot give any other reason except that it is so.

Reviewer 2: #8. Histological findings, including inflammatory cells infiltration, and hepatic fibrosis, should be described more in detail (Figure 4B).

We followed the advice of reviewer 2 and added a sentence in this section. We refer reviewer 2 to point #5 in this section and refer within the manuscript to the new point in the methods: The mHAI, rated as an average of 3, was composed of observed periportal interface hepatitis, apoptosis, and portal inflammation, whereas confluent necrosis was absent in both groups (line 238-240)

We would like to briefly point out that fibrosis can only be observed in this model from week 30 on. Therefore, unfortunately, fibrosis does not affect the results here either. Furthermore, fibrosis is not part of the mHAI.

Reviewer 2: #9. The fold change and p-value in each protein should be addressed (Figure 6). In Figure 6D, Wisp1 expression, which is up-regulated in the mice with fibrosis, was significantly decreased by the treatment with anti-CD20. But, no significant histological change was not found by the treatment. Moreover, IL-5 expression was enhanced by the treatment. The authors should mention the reasons. Tthe roles of Fst and Cntn1 should be also described more in detail.

We followed reviewers 3 advice and added the p-values to Figure 6.

The reviewer 2 mentioned that the manuscript did not explain why IL-5 and other proteins were differentially regulated. Unfortunately, however, the question goes far beyond the actual research question. We wanted to show that anti-CD20 has a positive effect on the course of emAIH. Since the therapy had no positive effect on the biochemical or histological parameters, we looked for explanations and found them in the differentially regulated proteome (e.g. IL-5, but not IL-6). Precise correlations between the proteins are work for another study by a group specialized in protein interactions. Unfortunately, this is clearly beyond our expertise and was not the aim of this study.

Reviewer 2: #10. I found the statement, “In our model, no regulation of IL-6 was observed (Line 291-292)”. But, its result was not shown in this study.

The reviewer 2 is correct, IL-6 was not mentioned in the text, but it was part of the 92 proteins that were analyzed. IL-6 was not differentially regulated. Therefore, our statement is correct.

Reviewer 2: #11. This study revealed that the treatment with anti-CD20 kept a balance between regeneration and inflammation in the model of AIH. But, the authors should describe how the treatment kept the balance more clearly.

The reviewer 2 mentioned an interesting point. Unfortunately, however, the question goes far beyond the actual research question. We wanted to show that anti-CD20 has a positive effect on the course of emAIH. Since the therapy had no positive effect on the biochemical or histological parameters, we looked for explanations and found them in the differentially regulated proteome. Precise correlations between the proteins are work for another study by a group specialized in protein interactions. Unfortunately, this is clearly beyond our expertise and was not the aim of this study.

Reviewer 2: Minor #1. “At week ten” should be corrected to “at ten week” (line 140).

We are sorry, but “at week ten” is (also) correct. The original manuscript was corrected by NatureSpringer editing service. We added the certificate.

Reviewer 2: #2. “Fig C,D.” should be corrected to “Fig.2C,D” (line 156).

We are sorry for this sloppy mistake and changed it.

Reviewer 2: #3. Sample number of the control group (n=4) in Figure 5 should be addressed.

We are not sure about this comment. We had four control mice for flow-cytometry analysis and mentioned this in the figure legend.

Reviewer 2: #4. The authors need to explain “q<0.05” (line 247).

We added a paragraph to the statistics in the methods section:

Alternatively, for multiple variables, the unpaired Student’s 2-tailed t-test with an implemented Benjamini-Hochberg multiplicity correction was performed using Qlucore Omics Explorer software 3.5 (Qlucore, Lund, Sweden). Heat map represents the multiple protein expression profiles (p<0.05; q<0.05). (line 149-153)

Reviewer 2: #5. “On the other hand” is duplicated (Lines 312-313).

 We are sorry, we could not find a duplication of this term.

References:

  1. Beland, K., et al., Depletion of B cells induces remission of autoimmune hepatitis in mice through reduced antigen presentation and help to T cells. Hepatology, 2015. 62(5): p. 1511-23.
  2. Hardtke-Wolenski, M., et al., Genetic predisposition and environmental danger signals initiate chronic autoimmune hepatitis driven by CD4+ T cells. Hepatology, 2013. 58(2): p. 718-28.
  3. Buitrago-Molina, L.E., et al., Treg-specific IL-2 therapy can reestablish intrahepatic immune regulation in autoimmune hepatitis. J Autoimmun, 2021. 117: p. 102591.
  4. Hardtke-Wolenski, M., et al., The influence of genetic predisposition and autoimmune hepatitis inducing antigens in disease development. J Autoimmun, 2017. 78: p. 39-45.

Reviewer 3 Report

Laura Elisa Buitrago-Molina and co-authors present a quality and well-written experimental manuscript that describes anti-CD20 alters the protein signature in experimental murine AIH but not exclusively towards regeneration.

Authors used a well-established model of experimental murine AIH (emAIH) and examined the effect of anti-CD20 treatment. They compared emAIH animals that received anti-CD20 treatment during the late course of disease with untreated controls.

Experimentally, authors used histopathology, biochemical parameters, intrahepatic and intrasplenic cellular components and activation status of the immune response. Also evaluated the signature of serum proteins that are involved in many different processes, such as angiogenesis, apoptosis, cell adhesion, differentiation, motility, proliferation, metabolic processes, chemotaxis, developmental processes, the immune response, the regulation of gene expression and the response to stress.

Finally, authors conclude that anti-CD20 treatment in AIH and emAIH remains an interesting tool for the treatment of the disease but is obviously not suitable as a monotherapy. They showed that this effect was due to the consequences of treatment at the protein level.

Other comments:

1) “Anti-CD20 antibody” can be used instead of “Anti-CD20” in the abstract and even the title - for the readers who are not necessarily experts in immunotherapy.

2) Cytokine release syndrome can be mentioned with regards to anti-CD20-associated increase of the pro-inflammatory protein levels.

3) Authors are kindly encouraged to cite the following article that overviews the application of adoptive immunotherapies. DOI: 10.3390/cancers13040743 (Pubmed ID: 33670139)

Overall, the manuscript is valuable for the scientific community and should be accepted for publication after minor edits are made.

Author Response

Reviewer3: Laura Elisa Buitrago-Molina and co-authors present a quality and well-written experimental manuscript that describes anti-CD20 alters the protein signature in experimental murine AIH but not exclusively towards regeneration.

Authors used a well-established model of experimental murine AIH (emAIH) and examined the effect of anti-CD20 treatment. They compared emAIH animals that received anti-CD20 treatment during the late course of disease with untreated controls.

Experimentally, authors used histopathology, biochemical parameters, intrahepatic and intrasplenic cellular components and activation status of the immune response. Also evaluated the signature of serum proteins that are involved in many different processes, such as angiogenesis, apoptosis, cell adhesion, differentiation, motility, proliferation, metabolic processes, chemotaxis, developmental processes, the immune response, the regulation of gene expression and the response to stress.

Finally, authors conclude that anti-CD20 treatment in AIH and emAIH remains an interesting tool for the treatment of the disease but is obviously not suitable as a monotherapy. They showed that this effect was due to the consequences of treatment at the protein level.

Other comments:

1) “Anti-CD20 antibody” can be used instead of “Anti-CD20” in the abstract and even the title - for the readers who are not necessarily experts in immunotherapy.

We thank Reviewer3 for this advice and have implemented it in the abstract. Nonetheless, we think it makes the title unnecessarily overbearing and would rather do without it at this point. Instead, we added “therapy” to the title. We hope this is in line with the reviewers 3 idea. Since readers will hopefully not only read the title, but also the abstract and the introduction, all ambiguities should be removed there. In the introduction we have used a (“hereafter referred to as anti-CD20”) to simplify the notation in the following manuscript.

Reviewer 3: 2) Cytokine release syndrome can be mentioned with regards to anti-CD20-associated increase of the pro-inflammatory protein levels.

We agree with the reviewer 3 and modified the manuscript:

At this point, it is reasonable to speculate about a connection between the cytokine release syndrome and rituximab, which is certainly known in certain circumstances [1]. (line 348-350)

Reviewer 3: 3) Authors are kindly encouraged to cite the following article that overviews the application of adoptive immunotherapies. DOI: 10.3390/cancers13040743 (Pubmed ID: 33670139)

We agree that this is a valuable reference and have added the manuscript to the bibliography.

Reviewer 3: Overall, the manuscript is valuable for the scientific community and should be accepted for publication after minor edits are made.

We would like to express our appreciation for the work of the reviewer 3 and for this assessment.

References:

  1. Bugelski, P.J., et al., Monoclonal antibody-induced cytokine-release syndrome. Expert Rev Clin Immunol, 2009. 5(5): p. 499-521.